# A Model to Evaluate the Effectiveness of the Maritime Shipping Risk Mitigation System by Entropy-Based Capability Degradation Analysis

**DOI:** 10.3390/ijerph19159338

**Published:** 2022-07-30

**Authors:** Jun Shen, Xiaoxue Ma, Weiliang Qiao

**Affiliations:** 1School of Maritime Economics and Management, Dalian Maritime University, Dalian 116026, China; shenjun15@mails.ucas.ac.cn (J.S.); maxx1020@dlmu.edu.cn (X.M.); 2Marine Engineering College, Dalian Maritime University, Dalian 116026, China

**Keywords:** navigational risk mitigation, effectiveness evaluation, degradation analysis, entropy of capability

## Abstract

Accurate evaluation of the risk mitigation status of navigating ships is essential for guaranteeing navigational safety. This research mainly focuses on the feasibility and accuracy of evaluating the real effectiveness of a risk mitigation system for navigating ships, including addressing the problem of immeasurableness for risk mitigation capability and determining the degradation regulation of risk mitigation capability over time. The proposed method to solve the problem is an effectiveness evaluation model based on the capability perspective, composed of a capability measurement algorithm based on entropy theory and capability degradation regulation analysis based on numerical process fitting. First, combined with the theoretical framework of a comprehensive defence system, the risk mitigation system designed for navigating ships is reconstructed based on capability building. Second, using a numerical fitting method, the degradation regulation of risk mitigation capability with time is obtained to improve the accuracy of the dynamic evaluation. Finally, referring to entropy theory, the uncertainty of capability is calculated, and then the model is constructed based on this uncertainty to realize the effectiveness evaluation from a capability perspective. The results obtained in an application test of the proposed model indicate that using the entropy of capability can realize an accurate effectiveness evaluation of a risk mitigation system for navigating ships, with a 9% improvement in accuracy, and the Weibull curve fitting is more consistent with capability degradation regulation, with a signification level of 2.5%. The proposed model provides a new path for evaluating the effectiveness of a risk mitigation system for navigating ships from the entropy of capability, and compared with the traditional probabilistic method, the model is more realistic and accurate in actual applications.

## 1. Introduction

### 1.1. Evaluation of Ship Risk Mitigation Status

With the rapid growth of the shipping industry and continuous changes in the navigational environment, the risk mitigation for navigating ships, as an important guarantee of navigation safety, has received more attention from the shipping industry and academia, especially the evaluation of the overall operation status of ship risk mitigation system (SRMS) under navigational state. Limited by the sailing scenario variability and time sequence degradation of risk mitigation capability, the traditional methods, including qualitative description and equipment function indicator measurement, cannot be accurately used to represent the real level of navigational safety [1]. Additionally, the risk mitigation status shows certain degradation characteristics (performance, adaptability, reliability) over time. Therefore, it is necessary to analyse and consider this problem in an evaluation of overall effectiveness.

As the main water transport means of international trade, the navigational safety of ships has always been the focus of shipping industry. In recent years, ship risk mitigation has gradually ushered in new challenges: on the one hand, ships are developing towards large-scale, high-speed, and intelligent uses, and the difficulty of ship risk mitigation has further increased; on the other hand, with the increase in traffic density for inland and coastal ships, the difficulty of ship navigational risk identification has further increased. Therefore, how to accurately evaluate risk mitigation status has become a core focus of ensuring ship navigation safety.

Similarly, with the development of navigation technology, risk mitigation techniques are being gradually interconnected to jointly complete risk mitigation action, showing some systematic characteristics. The interactions between these techniques and the resulting system uncertainty have gradually highlighted the systemic safety issues related to ship navigation. Therefore, in the evaluation process of ship risk mitigation status in the new era, the impact and role of ship technology participation need to be fully considered to solve systemic security problems on the basis of considering traditional human defence and physical defence.

Based on the above analysis, it is found that the current SRMS is a complex multidimensional system composed of human–ship interactions, ship–shore interactions, and ship–ship interactions, including crews, equipment, technology, management, and other elements. These elements are interrelated to form a comprehensive risk mitigation capability. Generated by the systematic development of ship risk mitigation measures, a comprehensive risk mitigation capability not only reduces the high consumption of traditional multipoint decentralized risk mitigation but also produces overall system effectiveness. Therefore, ensuring the comprehensive capability of SRMSs is particularly important for marine navigation safety. However, the application of various information technologies will inevitably result in new technological risks and operational uncertainty for navigating activities. For instance, the risk of network hijacking may be caused by the introduction of data transmission technology. Therefore, how to correctly identify the impact of these uncertainties on system capabilities and accurately evaluate the effectiveness of SRMSs has become a key issue to ensure the navigational safety of ships.

### 1.2. Related Work

With widespread concern about maritime safety around the world, a large number of evaluative studies on the status of ship risk mitigation have emerged in recent years. Since human factors have always been an important cause of ship accidents, most of the current research on the ship risk mitigation domain has focused on the assessment and analysis of the risk mitigation status of avoiding human error [2]. In practical applications, a quantitative method based on the analytic hierarchy process has been mainly used to analyse the failure indicators of ship personnel risk mitigation status [3,4]. This method has been successfully applied to human accident analysis, crew capability building, and crew comprehensive quality evaluation [5,6]. Additionally, in recent years, with the continuous development of science and technology, risk mitigation auxiliary technologies such as video surveillance, risk perception, and information communication in ship risk mitigation have also been widely applied [7,8,9]. The introduction of new technologies has improved the level of risk mitigation, but at the same time, it has also brought new security risks. Current research on the failure of new technologies and the risks associated with uncertainties has mainly used fuzzy comprehensive evaluations and Bayesian networks to evaluate and analyse the relevant performance indicators of SRMSs. Among them, the fuzzy comprehensive evaluation method has been successfully used for the safety assessment of ship security video monitoring, data interconnection, and packet loss [10,11,12]. However, the fuzzy comprehensive evaluation method is greatly affected by the subjectivity of the scorer and lacks a certain degree of objectivity in SRMS analysis. Similarly, the Bayesian network analysis method has also been successfully applied in the fields of risk mitigation strategy evaluation, risk mitigation system optimization, and risk mitigation information fusion [13,14,15]. However, this method relies on the regression analysis of past experience while evaluating the effectiveness of an SRMS and lacks applicability for the analysis of the relationship between discontinuous complex systems. In recent years, with the continuous enrichment of risk mitigation methods, the overall evaluation of SRMSs has gradually received attention in the navigation safety domain. Some scholars have carried out research on evaluating SRMSs from a multiple perspective and a full-chain perspective, focusing on analysing improvements to risk mitigation methods after an accident and evaluations of the cohesion between the risk mitigation links [16,17]. This “patch-style” optimization method has promoted the improvement of ship risk mitigation capability and the filling of the risk mitigation vacuum. However, a navigating ship is faced with complex and changeable external environments and various uncertain challenges all the time, so the “patch” can never be completed, and this method cannot fundamentally solve the problem. Therefore, a capability construction method for building a “universal” system for ship risk mitigation is more effective than a traditional “patch” scattered risk mitigation method. In a recent research report, it was proposed to analyse the relationship between the composition of the system’s capabilities and the performance of the system [18]. On this basis, the capability measurement method was used to calculate the effectiveness of a complex system [19]. However, this method equates effectiveness with capability and does not fully consider the time-series system capability change attributes and navigation scenario change attributes of SRMSs, which affects the feasibility and accuracy of the evaluation.

Based on previous research, this paper proposes a new effectiveness evaluation methodology for SRMSs, including entropy-based capability calculation and capability degradation analysis. The methodology considers a system’s scenario variability and time sequence degradation. A quasi-normal distribution of statistical parameters is combined with a Weibull distribution based on degradation analysis theory to achieve an accurate effectiveness evaluation. The improved calculation method of capability uncertainty constructed by entropy theory is used to address the problem of capability mismeasurement. The contribution of this study is summarized as follows.

(1)Exploration in entropy-based model to investigate the effectiveness of the risk mitigation system when defencing various risks, the results of which are able to benefit greatly the safety management.(2)Establishment a complete solution for the effectiveness assessment model for the ship risk mitigation system, which can be used to improve the risk management of the maritime transportation.(3)The performance of the proposed model is verified by a case study, which indicates the great application potential in the field of risk management.

### 1.3. Organization

The remainder of this paper is organized as follows. Section 2 presents an overview of the research methodology and the employed techniques. Section 3 presents the model establishment process. Section 4 verifies the proposed model with an actual application case. Section 5 concludes the study and presents future directions. Additionally, an explanation of all the collection data used in this study is given in Appendix A.

## 2. Methodology

An overview of the methodology proposed in this study is illustrated in Figure 1, in which the integration of a comprehensive security system theory, a capability degradation analysis method, and the entropy of capability leads to the formation of a comprehensive model to analyse the risk mitigation status of ships under navigation. As shown in Figure 1, there are 3 parts to this study:

Part 1—evaluation framework construction. An evaluation indicator framework is obtained based on structure reorganization using a comprehensive defence framework.

Part 2—indicator degradation analysis. The capability degradation regulation is obtained through numerical curve fitting.

Part 3—system effectiveness evaluation. The system effectiveness is obtained by calculating the degree of capability uncertainty by entropy theory.

### 2.1. Comprehensive Defence System

The comprehensive defence system provides theoretical support for the reorganization of SRMSs from a traditional decentralized prevention framework based on risk mitigation subject to the capability building framework based on risk mitigation tasks. The comprehensive defence system originated from the physical protection system (PPS), which was first proposed by Sandia Laboratories in the 1980s and was widely used in the field of nuclear safety protection [20]. For the first time, this theory proposes a “triad” safety protection theoretical framework composed of humans, material objects, and technology from the perspective of improving overall risk mitigation capabilities [21]. The principle is shown in Figure 2.

In Figure 2, the theoretical framework of risk mitigation emphasizes the coordination, interdependence, complementarity, and cooperation of multiple risk prevention and control methods. Different from the traditional ergodic risk-responsive safety protection theory, this theory emphasizes the diversification of risk mitigation methods, the systematization of risk mitigation organizations, and the integrity of risk mitigation capabilities and strives to transform the construction of discrete risk mitigation points into the construction of integrated capability. It is more advantageous to use the risk mitigation framework based on comprehensive capability building to address changeable risks in a complex navigational environment, especially in terms of flexible response and timely control.

### 2.2. Capability Degradation Analysis

The capability degradation analysis method provides an effective means for quantitatively grasping the regulation of system capability changing with time. With the basic understanding that an SRMS has a prescribed life cycle, we infer that the performance of its system components is continuously degraded with time, which will lead to a degradation of the system capability. Therefore, this study adopts the capability degradation analysis method to analyse the degradation regulation of the component units, which can accurately identify the dynamic characteristics and real-time state of risk mitigation capability [22].

#### 2.2.1. Theoretical Basis of Degradation Analysis

Degradation theory is the core foundation of degradation analysis and the main theoretical basis for the exploration of degradation law. Degradation theory was first put forward in biology, which mainly refers to the transformation trend and process of material system from order to disorder or from high order to low order in natural evolution and is used to express the analysis of the continuous decline degree of biological performance. The main core of degradation theory is that as time goes on, a certain function of anything will gradually decay. This is consistent with the law of our real world. The introduction of degradation theory makes it possible to master the law of performance change with time. On the one hand, for many devices in our research, degradation is a natural attribute. Whether there is failure or not, their performance data can be monitored to get degradation data. On the other hand, due to the modelling of degradation data closer to the results of failure physics, the introduction of degradation theory can provide more information for the degradation process, and the degradation process can also be used to find the correlation between equipment performance degradation and system state.

#### 2.2.2. Degradation Form Judgment and Indicator Selection

Effective degradation form judgement is the basis for analysing the regulation of ship risk mitigation capability change. According to the status of a component unit losing its prescribed function, the failure of a component unit in SRMS can be divided into two forms: sudden failure and degenerative failure. Due to inevitable sudden failures in actual risk mitigation, the form of sudden failure needs to be removed in the degradation analysis to reduce its interference with the analysis of routine degradation regulation.

Unlike sudden failures, degenerative failures refer to a slow decline in performance over time. It is usually represented by a degradation function that changes with time. Considering that the performance of component units affecting the system risk mitigation capability needs to reserve a certain margin of insurance, the performance standard to ensure the normal operation of the system is set as a fixed value R, and its corresponding equipment effective life threshold is T, as shown in Figure 3:

The accurate selection of the degradation indicator is a core task for mastering the degradation regulation of ship risk mitigation capability. Clarifying the characteristic indicators and failure criterion of a component unit is the basic goal when analysing the system effectiveness. The performance characteristic quantity of a component unit is a parameter value reflecting the capability change of a risk mitigation subject, and it needs to be selected according to the function setting by a specific risk mitigation task [23]. In this study, the criterion was usually determined according to the requirements of risk mitigation tasks and called the failure threshold [24]. The failure threshold can be a determined value or an interval.

The selection of performance failure characteristic quantity and data acquisition usually have the following considerations: (1) The performance failure characteristic quantity can be an output parameter specified by the risk mitigation subject [25]. (2) Measurability is the basic requirement for the setting of the failure characteristic quantity. (3) Considering the regularity of measurement data and the convenience of experimental development, in this study, the time interval of each sample measurement is set to be the same.

#### 2.2.3. Degradation Regulation Analysis

After selecting the corresponding performance characteristic quantity, exploring the capability degradation regulation of a component unit becomes the top priority. According to the natural degradation phenomenon, the component unit of an SRMS can be classified into the loss component with time. Based on reliability theory, the data distribution of the failure characteristic quantity can be approximated as subject to some known empirical distribution functions, such as normal, exponential and Weibull distributions [26]. Combined with probability theory, the failure probability at a certain moment can be obtained through the distribution function of the failure characteristic quantity at the moment to finally realize the effectiveness evaluation of the SRMS. The degradation regulation analysis can be roughly divided into the following two steps:

The first step is extracting the true value of the failure characteristic quantity. The main goal is to process the corresponding measurement data by mathematical statistics on the basis of the above selected methods. Assuming that the component unit stays in the normal state, the failure degree of its performance at a certain point can be determined. However, due to the existence of measurement error, there will be some differences between the measured value and the real value. Therefore, the values of multiple measured samples at the measurement point obey the normal distribution form as follows:(1)$$x\sim N\left(\mu ,\sigma^{2}\right)$$
where x represents the measured value, μ represents the true value, and *σ* represents the standard deviation of the measurement, as shown in Figure 4.

The second step is the numerical curve fitting of the characteristic quantity. An appropriate distribution function is typically selected to fit the curve of the characteristic variable with time. Since assuming that the risk mitigation unit follows the regulation of capability degradation, its characteristic value should also follow the corresponding degradation regulation. By comparing and analysing the exponential, logarithmic, and Weibull forms, the optimized distribution form can be selected to fit the corresponding failure characteristic values, as shown in Figure 5.

### 2.3. Entropy of Capability Calculation

The method of entropy measurement provides an effective means for quantitatively analysing the indirect measured quantity of capability. Based on the basic belief that a change in risk mitigation status for navigating ships is equivalent to a change in risk mitigation effectiveness caused by the uncertainty of a system’s risk mitigation capability, the effectiveness value can be obtained by measuring the uncertainty of risk mitigation capability [27]. This approach fundamentally addresses the difficult problem that risk mitigation capability cannot be measured.

According to the concept of information entropy in entropy theory designed for measuring the uncertainty of an information source, the measurement of information is characterized by the uncertainty of the information source [28]. By analogy, the method of entropy of capability can be constructed as the uncertainty measure of a capability source, that is, the measure of capability is characterized by the uncertainty of its component units. Based on this analogy, it is assumed that the component factor of a certain type of capability is $A_{i}$. The influence degree of factors on this capability is $p_{i}$, expressed by the basic probability of causing capability failure. Therefore, the uncertainty degree $H_{i}$ caused by this factor to the capability can be expressed by the self-information quantity function:(2)$$H_{i}=ln\frac{1}{p_{i}}$$

Since the sum of $p_{i}$ in traditional entropy theory is 1, the measurement of information will show a convergence trend [29]. Therefore, the uncertainty of the system is the uncertainty of the value of the system factors. Assuming that there are n factors that affect the protection capability, according to the nature of the entropy function in information theory, when $p_{i}=\frac{1}{n}$, the entropy value is the largest [30]. However, the result of this assumption is inconsistent with the objective understanding of protection effectiveness in practice. In reality, the smaller the failure probability of the factor affecting the protection capability, the smaller the instability it brings to the protection ability, the greater the protection capability, and the higher the protection effectiveness. Therefore, the traditional concept of information entropy cannot be directly used here as an analogy for risk mitigation effectiveness.

Considering that the capability of SRMS is affected by the component units and has a positive correlation, the entropy of capability in the SRMS is not the probability of the system capability value but the strength of the uncertainty of conduction correlation between the units and the system. Therefore, the entropy of capability fundamentally solves the uncertainty of the capability fluctuation rather than the uncertainty of the capability value. On this basis, the entropy of capability based on conduction correlation is constructed as follows:(3)$$I_{i}=(1-p_{i})ln\frac{1}{p_{i}}$$

Furthermore, it can be concluded that the entropy of capability does not have the characteristics of convergence, which is consistent with reality. Therefore, the greater the uncertainty of a system risk mitigation capability, the greater the uncertainty of its system risk mitigation effectiveness, and then the greater the entropy of capability.

## 3. Model Establishment

### 3.1. Framework Reorganization of SRMS

#### 3.1.1. System Analysis and Reorganization

With the continuous enrichment of risk mitigation means and technology, the composition of SRMSs is constantly deepening and developing, from single components to integrated subsystems and then to the current comprehensive risk mitigation system.

Most of the common SRMS frameworks have been designed based on the spatial distribution of risk mitigation points, mainly including bridges, engine rooms, cargo holds, decks, and living areas [8]. Each risk mitigation point is composed of multiple elements, as shown in Figure 6. The element details are as follows:The bridge subsystem includes elements such as navigation operators, navigation monitoring and warning systems, and navigation communication equipment.The engine room subsystem includes elements such as marine engineers, engine room monitoring platforms, and engine room maintenance equipment.The cargo hold subsystem includes elements such as operators, monitoring systems, and emergency equipment.The deck subsystem mainly includes elements such as staff, protection equipment and ship rescue equipment.The living cabin mainly includes elements such as personal protective equipment and personnel health protection systems.

Due to continuous changes in the navigation environments of ships and the continuous improvement of shipping safety requirements, the complexity and suddenness of the current ship risk mitigation situation has become increasingly serious [31]. The risk mitigation effectiveness of this common SRMS adopting single-point strict prevention means is worsening. Combined with the capability system architecture design of human, physical and technical defences proposed in the comprehensive defence system, the multiple elements in the five subsystems are classified and reorganized, as shown in Figure 6. The system forms the following three parts:Risk mitigation methods, including human-based behaviours, strategies, and measures.Risk mitigation equipment, including hardware-based equipment, signs, and facilities.Risk mitigation platforms, including technology-based perception models, monitoring software, and early warning systems.

Based on the risk mitigation capability construction, the reorganized framework of the SRMS has a clear boundary in structural decomposition and a relatively independent carrier in capability measurement. It provides a practical system network architecture for effectively mastering and analysing the actual effectiveness of SRMSs.

#### 3.1.2. System Description

The component units of SRMS are independent of each other in the risk mitigation carrier, but they are related to each other in terms of function. Through interaction, a comprehensive capability of risk mitigation is jointly formed. According to the composition, functions and characteristics of each subsystem in the SRMS, its basic architecture and constituent units are analysed and integrated as follows:(1)Human defence subsystem

Human defence is the oldest method in the risk mitigation domain, and it is also an indispensable protective measure used since ancient times. In the risk mitigation system, human defence mainly includes all related personnel involved in the risk mitigation subjects [32]. Its core role is to use human sensors (eyes, hands, ears, etc.) for detection. When an attacker is found to be dangerous to the protection target, it will rely on its own identification or information transmission to obtain risk signals, make judgements about it and then take corresponding measures. Based on this understanding, the human defence subsystem of the SRMS can be defined as a collection of artificial behaviours for risk mitigation established by the risk mitigation awareness, consciousness and capability of the behavioural associates to realize ship navigation safety. Its basic composition, objectives, functions and characteristics are shown in Table 1.

In Table 1, one of the cores of the risk mitigation system established for navigating ships is humans. Therefore, the related organizations and personnel involved in ships need to put humans in the first place, and education on risk mitigation is essential, covering all links focusing on the education and training of personnel involved in risk mitigation awareness, knowledge, and skills. Additionally, there must be a focus on improving human risk mitigation capability, especially allowing ship managers, operators, and risk mitigation specialists who participate in actual risk mitigation strengthen safety responsibilities ideologically to improve professional ethics in concepts and develop risk mitigation habits in terms of behaviour. To strengthen human defence, the key is to improve and implement the responsibility system for risk mitigation, clarify the functions and responsibilities of all personnel, and strengthen the supervision, inspection and assessment of the implementation of the risk mitigation responsibility system.

Based on a statistical analysis of ship accident reports and on-site investigations and research, combined with the above definition of the human defence subsystem, we divide it into the following units:Ship management unit. This is mainly composed of ship owners and ship controllers. It is responsible for the maintenance plan, personnel arrangement, financial support and other aspects of ship risk mitigation. This is the top-level design unit of the human defence subsystem of the SRMS.Post operation unit. This is mainly composed of operators and supplementary personnel in various positions, such as the bridge, engine room, and deck of the ship. They are responsible for the safe operation of specific positions and the handling and response of direct risks under the ship’s sailing state. This is the core response unit of the human defence subsystem of the SRMS.Shore-based assistance unit. This is mainly composed of shore-based ship dispatching, supervision and piloting personnel. It is responsible for real-time monitoring, regular inspections, and assistance in response to the ship navigational risk state from the shore. This is an important guaranteed part of the human defence subsystem of SRMS.

(2)Physical defence subsystem

Physical defence is the core of early systematic protection, and it is also one of the oldest risk mitigation methods used to deal with safety risk. In the risk mitigation system, the physical defence subsystem generally refers to the hardware, physical objects and other barriers that protect the safety of the object [33]. Based on this, the physical defence subsystem of an SRMS can be defined as a collection of physical risk mitigation objects relying on the attributes, configuration and functions of the physical risk mitigation system to realize ship navigation safety. Its main risk mitigation objects include installing protections, facilities, and tools while strengthening protective barriers in key parts, important areas, and key places; setting up hardware facilities such as fire rescue and safe operation; equipping the post personnel with necessary duty, protection, inspection, and other equipment; and setting signs such as prompt board, safety board and restricted access signs. Its basic composition, objectives, functions and characteristics are shown in Table 2.

In Table 2, one of the cores of the safety risk physical defence subsystem established for ships in navigation is hardware. This means that the related parties of ship safety navigation need to increase the resource investment of hardware to provide strong hardware guarantees for the smooth development of safety protection and management of ships during a voyage. It is also an important material basis for building a comprehensive risk mitigation system for safety risk responses. On the premise that the configuration of risk mitigation hardware covers all links of the whole shipping chain, the function and combination of risk mitigation hardware shall be determined according to the actual risk mitigation task requirements. The maintenance and updating of risk mitigation hardware require the support of corresponding funds. Under the overall requirements of reducing ship operation costs, relevant departments related to navigational safety must scientifically plan the budget of hardware funds related to ship navigation safety to effectively ensure the normal and orderly operation of the physical defence subsystem.

Based on a statistical analysis of ship accident reports and on-site investigation and research, combined with the above definition of a physical defence subsystem, we divide it into the following units:Safety facility unit. This is mainly composed of protective equipment related to navigation safety and ship fire protection. It is the basic unit of passive ship risk mitigation.Prevention barrier unit. This is mainly composed of a safety valve, protective net, and explosion-proof door to delay and hinder risk diffusion. It is a supplementary unit for passive risk mitigation.Prompt identification unit. This is mainly composed of indicative signs such as reminder boards, safety boards, and restricted access signs set up at a fixed position. It is an important unit for ship safety protection.Personnel equipment unit. This is mainly composed of the necessary protection and inspection equipment for post personnel. It is the basic material guarantee to support post personnel to effectively deal with navigational risk.

(3)Technical defence subsystem

Technical defence is a new risk mitigation method in the information age and is the main technological core of the current risk mitigation system. Technical defence refers to protection means using modern science and technology, especially modern information technology means, with strong technical characteristics such as various detection systems, early warning and alarm systems, video monitoring systems, and access control systems [34]. This is an extension and strengthening of human defence and physical defence in protection means and the supplementation and improvement of human defence and physical defence. Based on this, the technical defence subsystem can be defined as a collection of risk mitigation technologies for navigational ships with the characteristics of digitization, networking and intelligence, which rely on emerging technologies in the fields of sensing, computing, communication and processing.

Technical defence subsystems mainly refer to the design and development of corresponding information platforms and application software based on computer communication technology according to the actual risk mitigation requirements to realize the corresponding risk mitigation functions. When it is found that an attack object is dangerous to the protection target, the technical defence subsystem relies on its own information perception or processing capability to obtain potential hazard signals, effectively compensating for the inefficiency of the human defence subsystem and the solidification of the physical defence subsystem. Its basic composition, objectives, functions and characteristics are shown in Table 3.

In Table 3, one of the cores of the technical defence subsystem for ship navigational risk is the information processing system. With the rapid development of modern information technology, it is being widely used in the process of storing, processing and transmitting information in SRMSs. Especially in the construction of ship navigation information platforms, the construction of information systems and the construction of safety risk technology protection systems should be synchronously planned, designed, and constructed. The production, transmission and use of vessel navigation information and its carriers must be equipped with technical equipment that meets the standards for information safety. Key positions for risk mitigation must take measures to prevent and eliminate hazards and be equipped with perfect technical equipment.

Based on a statistical analysis of ship accident reports and on-site investigation and research, combined with the above definition of a technical defence subsystem, we divide it into the following units:Navigation monitoring unit. This is mainly composed of modern information protection platforms such as bridge information monitoring, radar monitoring, and weather monitoring. It is the basic functional unit for the ship technical risk mitigation subsystem.Information assurance unit. This is mainly composed of information technology protection methods such as emergency communication platforms, network protection means, and ship–shore cooperative communication guarantees. It is the basic guarantee unit for the ship technical risk mitigation subsystem.Risk warning unit. This is mainly composed of specific risk mitigation technologies such as the identification of unsafe behaviours, the alarm of abnormal routes, and the fault tolerance of the warning system. It is the core application unit for the ship technical risk mitigation subsystem.Decision management unit. This is mainly composed of the specific safety risk treatment means of automatic collision avoidance route planning, ship automatic navigation and safety risk autonomous response systems. It is an important response unit for the realization of ship technical risk mitigation.

### 3.2. Indicator Design of SRMS

Based on the human defence, physical defence, and technical defence framework architecture in the above comprehensive defence system and considering the transformation of the risk mitigation concept from discrete risk response to system capability construction in SRMSs, a comprehensive capability indicator framework is proposed in this study, combined with the actual understanding of SRMSs.

#### 3.2.1. Human Defence Subsystem

Combined with human protection characteristics such as human initiative and the flexibility proposed in system safety protection, the human defence subsystem is actually a collection of risk mitigation standards, measures and behaviours that realize ship navigation safety, relying on the risk mitigation consciousness, consciousness and capability of behaviour-related persons and human subjects [35]. Based on an analysis of SRMS structure and the compilation of questionnaires for on-board personnel, a framework for evaluating the capability of a human defence subsystem is constructed. It is mainly divided into three parts: human-oriented attributes, responsibility planning, and system guarantees, as shown in Figure 7.

In Figure 7, human-oriented attributes are used to measure whether the basic capability attributes of associated personnel meet the needs of risk mitigation capabilities, mainly including skill level (age, experience, training, communication, emergency), safety awareness (awareness of compliance with regulations, self-protection, and group cooperation), and personnel health (physical and psychological).

In Figure 7, responsibility planning is used to measure whether the manpower allocation of the human defence subsystem meets the needs of risk mitigation capability, mainly including whether the distribution of power and responsibility is reasonable (rationality of post setting, applicability of staffing requirements, accuracy of staffing quantity), whether job training is in place (pertinence of training content, compliance rate of training frequency, accuracy of training objects), and post-assessment accuracy (assessment content, assessment method and assessment result).

In Figure 7, system support is used to measure whether the support force meets the requirements of supporting the human defence system to give full play to the risk mitigation ability, mainly including team building (personnel age structure, personnel education structure, personnel salary structure), incentive mechanisms (whether the incentive strength is enough and the incentive scale is accurate), and material support (living materials, work materials and emergency materials).

Based on the abovementioned human defence subsystem measurement framework and considering the ergodicity and effectiveness of capability indicators, we selected 11 sets of core capability indicators from historical data and risk mitigation node analysis and used them to ensure measurability. On the above, measurable indicators are set for the corresponding indicators, as shown in Table 4.

#### 3.2.2. Physical Defence Subsystem

The physical defence subsystem is regarded as object protection in safety protection. It is a collection of instruments, signs and facilities for ship risk mitigation with hardware as the main body for the purpose of realizing ship navigation safety, giving full play to the passive damage resistance of objects to reduce and delay potential risk [36]. Based on the trace-to-source safety risk and investigation of ship-related personnel, a framework for evaluating the capability of the physical defence subsystem is constructed, which is mainly divided into three parts: entity attributes, combination configurations, and system guarantees, as shown in Figure 8.

In Figure 8, entity attributes are mainly used to measure whether the basic performance quality of the entity meets the requirements of physical defence, mainly including core function (damage resistance, replaceability) and basic quality (service life, consistency and failure rate).

In Figure 8, the combined configuration is mainly used to measure whether the physical configuration of the physical prevention subsystem meets the requirements of risk mitigation capability, mainly including physical selection (the rationality of physical risk mitigation settings, the applicability of physical allocation requirements, and the accuracy of physical configuration quantity), physical response design (whether physical use training is available, and whether the cooperation between physical objects is good), and the actual effect feedback of the material object (feedback content, feedback method and effect after feedback).

In Figure 8, system support is mainly used to measure whether the support force can meet the requirements of supporting the risk mitigation capability of the physical defence system, mainly including physical maintenance (whether the maintenance frequency is enough, whether the maintenance personnel are good enough, whether the maintenance content is accurate), quality control (whether the acceptance control of equipment is not controlled, whether the hidden danger of equipment is not dealt with, whether the operation environment is optimized or not), and funding guarantee (equipment purchases, equipment maintenance, upgrade replacements).

Based on the abovementioned physical defence subsystem measurement framework and considering the ergodicity and effectiveness of capability indicators, we selected 13 sets of core capability indicators from historical data and risk mitigation node analysis and used them to ensure measurability. On the above, measurable indicators are set for the corresponding indicators, as shown in Table 5.

#### 3.2.3. Technical Defence Subsystem

Technical defence is regarded as the extension and enhancement of human defence and physical defence in risk mitigation. It is a collection of risk mitigation methods, application software and integrated systems with technology as the main body for the purpose of realizing ship navigation safety, giving full play to the effectiveness of technical activities from the perspective of technical prevention, using information technology to control, discover, analyse and deal with potential risk. Based on the trace to source of safety risk and investigation of ship-related personnel, a framework for evaluating the capability of technical defence subsystem is constructed, which is mainly divided into three parts: an ontology attribute, function setting, and system guarantee, as shown in Figure 9.

In Figure 9, the ontology attribute is used to measure whether the basic function module design of the system meets the calibre requirements of technical defence, mainly including system functionality (information integration and function expansion), system reliability (environmental adaptability, compatibility between systems), and system portability.

In Figure 9, the function setting is used to measure whether the configuration of the technical defence system meets the requirements of risk mitigation capability, mainly including the integrity of the system function (coverage, target type, monitoring accuracy probability), accuracy (measurement accuracy of technical defence, information fusion accuracy, communication efficiency) and timeliness.

In Figure 9, the system support is used to measure whether the support force meets the requirements of the support defence system to give full play to the risk mitigation capability, mainly including maintenance mechanism (whether the maintenance frequency is enough, whether the maintenance personnel can do it, and whether the maintenance content is accurate), emergency mechanism (emergency response to system failure), and rescue mechanism.

Based on the abovementioned technical defence subsystem measurement framework and considering the ergodicity and effectiveness of capability indicators, we selected 12 sets of core capability indicators from historical data and risk mitigation node analysis and used them on the basis of ensuring measurability. On the above, measurable indicators are set for the corresponding indicators, as shown in Table 6.

### 3.3. Capability Degradation Analysis at Subsystem Level

Based on the above characteristic parameter setting and degradation regulation analysis and considering the scene volatility and time series degradation characteristics of SRMSs, this study proposes a capability degradation methodology based on failure characteristic quantity analysis. It is mainly divided into the following two parts:

#### 3.3.1. Parameters Characterizing Degradation Process

According to mathematical statistics theory, the characteristic parameter data should be obtained through a certain measurement experiment, and the measured data should be recorded at each time point in time order in the specific test [37]. To reduce the single measurement error of the component unit, the measurement of multiple similar samples is selected at one time point instead of multipoint measurement for a single sample at the same time point. Additionally, a certain number of test points are selected in the rated life cycle partition interval to reduce test numbers.

Assuming that a component unit of the SRMS is selected as a sample for the capability degradation analysis, the degradation characteristic parameter data of the corresponding unit are recorded in chronological order at each time point. For the objects of evaluation sample set $A=\left\{1,2,\ldots ,\;i,n-1,n\right\}$, we also create indicators of the evaluation sample set $B=\left\{1,2,\ldots ,\;j,m-1,m\right\}.$ Considering the different types of measurement data, the numerical polarization technique is introduced to normalize the degradation characteristic parameter [38]. Then, under the assumption that $x_{j}^{i}\left(t\right)$ represents the value of the jth indicators expressed for the ith evaluation object at the test time point t, we can obtain the evaluation sample result as follows:(4)$$\left\{x_{j}^{i}\left(t\right)|i\in A,j\in B\right\}$$
(5)$$x_{j}^{i}\left(t\right)\sim \left(0,1\right)$$

Due to the interference of equipment factors, measurement errors will inevitably exist in the test. Assuming that the theoretical value for this kind of sample at a certain time is $z_{j}\left(t\right)$ and according to the theory of quasi-normal distribution of statistical parameters, it is also represented as $y_{j}\left(t\right)$ by the expected value comparison method; thus, we can obtain:(6)$$x_{j}^{i}\left(t\right)=z_{j}\left(t\right)+\varepsilon_{ij}=y_{j}\left(t\right)+\varepsilon_{j}^{i}\left(t\right),\;\varepsilon_{j}^{i}\left(t\right)\sim \left(0,\sigma^{2}\right)$$
(7)$$y_{j}\left(t<t_{1}\right)\leq y_{j}\left(t<t_{2}\right),t_{1}\leq t_{2}$$
where $\varepsilon_{j}^{i}\left(t\right)$ represents the measurement error value of the jth indicators expressed for the ith evaluation object at the test time point t.

#### 3.3.2. Degradation Regulation Analysis

The purpose of building the degradation analysis method is to construct a suitable curve function to fit the trend of degradation characteristic parameters over time. It is assumed that the SRMS follows the capability degradation regulation, so its effectiveness characteristic value should also follow the corresponding degradation regulation. Combined with the theory of equipment reliability analysis and life tests, the optimized Weibull distribution is selected to fit the degradation characteristic curve [39]. The statistical value of the degradation characteristic parameters corresponds to a Weibull distribution, so its probability density function can be expressed as:(8)$$f\left(x;\lambda ,\kappa \right)=\frac{\kappa }{\lambda }{\left(\frac{x}{\lambda }\right)}^{\kappa -1}\times e^{-{\left(\frac{x}{\lambda }\right)}^{\kappa }}$$
where x represents the probability of the effectiveness characteristic value, λ represents the scale parameter, and κ represents the shape parameter.

Applying Equation (11), it can be determined that the value of $y_{j}\left(t\right)$ follows the optimized Weibull distribution. Additionally, the effectiveness characteristic value is expressed as the effective life value of the system component unit, so its change function can be represented as the probability density function $f\left(x;\lambda ,\kappa \right)$. On this basis, the trend of degradation characteristic parameters over time can be calculated as:(9)$$y_{j}\left(t\right)=f_{j}\left(t;\lambda ,\kappa \right)=\frac{\kappa }{\lambda }{\left(\frac{t}{\lambda }\right)}^{\kappa -1}\times e^{-{\left(\frac{t}{\lambda }\right)}^{\kappa }}$$

Then, based on the expected value of measurement $y_{j}\left(t\right)$ and fitting curve function $f_{j}\left(t;\lambda ,\kappa \right)$, the corresponding scale parameter λ and shape parameter κ can be obtained by statistical analysis and calculation of the maximum similarity value.

### 3.4. System Effectiveness Measurement Based on Entropy of Capability

#### 3.4.1. Measurement Methodology Description

The effectiveness measurement methodology based on entropy theory provides an effective theoretical basis for solving the accurate measurement of the actual status of ship risk mitigation. The specific derivation logic is shown in Figure 10.

In Figure 10, it can be seen that to realize the accurate measurement of the actual effect of ship risk mitigation, the traditional evaluation method of ship risk mitigation capability is faced with the inherent dilemma that the capability cannot be measured directly [40]. In addition, the specific application scenario characteristics of the SRMS cannot be considered in the evaluation of capability, so the actual status of risk mitigation under the specific task scenario cannot be truly reflected in it. Therefore, combined with the effectiveness definition of the completion degree of the capability to specific tasks, the measurement of the actual status of ship risk mitigation can be transformed into the solution of ship risk mitigation effectiveness.

However, according to the probability attribute characteristics of the capability to complete the task in effectiveness, capability is not measurable and therefore not probabilistic. Considering this, we assume that the initial ability can fully complete the task requirements, but the uncertainty of the capability occurs due to the scenario change, which leads to the change of the degree of capability to complete the task.

According to the aforementioned principle, the measurement of risk mitigation effectiveness can be converted into the uncertainty measurement of risk mitigation capability.

Combined with the definition and formula of entropy of capability in Section 2.3, the effectiveness measurement equation can be constructed as follows:(10)$$M_{i}=I_{i}=(1-p_{i})ln\frac{1}{p_{i}}$$
where $p_{i}$ represents the failure characteristic value of the ith element affecting capability.

In addition, according to capability degradation Equation (12) in Section 3.3, the time-series failure probability measurement value of the system components can be calculated as follows:(11)$$p_{i}=\frac{F_{i}\left(t\right)}{F_{fixed}\left(t\right)}=\frac{f_{i}\left(t;\lambda ,\kappa \right)=\frac{\kappa }{\lambda }{\left(\frac{t}{\lambda }\right)}^{\kappa -1}\times e^{-{\left(\frac{t}{\lambda }\right)}^{\kappa }}}{T_{fixed}}$$
where $T_{fixed}$ represents the rated service life value.

Therefore, the calculation equation of the system risk mitigation effectiveness based on capability degradation can be obtained as follows:(12)$$M_{i}=\left(\frac{f_{i}\left(t;\lambda ,\kappa \right)=\frac{\kappa }{\lambda }{\left(\frac{t}{\lambda }\right)}^{\kappa -1}\times e^{-{\left(\frac{t}{\lambda }\right)}^{\kappa }}}{T_{fixed}}-1\right)\;ln\frac{f_{i}\left(t;\lambda ,\kappa \right)=\frac{\kappa }{\lambda }{\left(\frac{t}{\lambda }\right)}^{\kappa -1}\times e^{-{\left(\frac{t}{\lambda }\right)}^{\kappa }}}{T_{fixed}}$$

#### 3.4.2. Comprehensive Effectiveness Aggregation

Based on the effectiveness measurement framework of “performance–capability–effectiveness” and combined with the above capability degradation methodology, the effectiveness measurement process is systematically integrated [41]. It is mainly divided into the following steps:(1)Indicator data pre-processing

Based on the indicator design of the SRMS, it is assumed that the expected value of each indicator is the characteristic quantity of the constituent elements affecting the capability. According to the rated performance requirements of constituent elements, the classification indicator value of each indicator is set $y_{low}$ as unqualified, $y_{mid}$ as qualified, and $y_{hign}$ as excellent. Combined with the algorithm of data extremum optimization, performance characteristic quantity is carried out with data cleaning. Therefore, the improved performance characteristic quantity can be obtained as follows:(13)$$s_{i}=0.8+\frac{y_{i}-y_{mid}}{y_{high}-y_{low}}$$
where $y_{i}$ represents the performance characteristic value of the ith component unit and $s_{i}$ represents the improved performance characteristic value of the ith component unit.

Then, a logical judgement is made for the performance characteristic indicator under each capability, and it is determined whether to use the following parallel algorithm or series algorithm for fusion:(14)$$\mathrm{parallel}\;\mathrm{algorithm}:\;q_{y}=\overset{n}{\underset{i=1}{\prod }}s_{i},\;\mathrm{series}\;\mathrm{algorithm}:\;q_{y}=\overset{n}{\underset{i=1}{\sum }}\omega_{i}s_{i}$$
where $q_{y}$ represents the performance characteristic fusion value of the yth capability.

(2)Effectiveness calculation based on entropy of capability

Combined with the performance measurement methodology constructed in Section 3.4.1, the probability value of the characteristics is transformed into a single performance value as follows:(15)$$e_{y}=(1-q_{y})ln\frac{1}{q_{y}}$$
where $e_{y}$ represents the effectiveness of the yth capability.

Then, the unit effectiveness value under the parallel algorithm can be calculated as:(16)$$e_{y}=(1-q_{y})\ln\frac{1}{q_{y}}=-\left(1-\overset{n}{\underset{i=1}{\prod }}\left(0.8+\frac{y_{i}-y_{mid}}{y_{high}-y_{low}}\right)\right)\ln\overset{n}{\underset{i=1}{\prod }}\left(0.8+\frac{y_{i}-y_{mid}}{y_{high}-y_{low}}\right)$$

(3)Subsystem performance integration

To facilitate the analysis and comparison of each system, based on the indicator system framework constructed above, this study separately calculates the human, physical, and technical defences and then integrates the three subsystems as follows:(17)$$I_{y}=e_{y}*\omega_{y},\;\omega_{y}=\frac{v_{y}}{\sum_{1}^{n}v_{y}}$$
where $\omega_{y}$ represents the loss value of risk accidents caused by the yth capability in the past 10 years, and $v_{y}$ represents the loss value caused by the yth capability under safety out of control.

Then, the jth subsystem efficiency value is
(18)$$E_{j}=\overset{n}{\underset{y=1}{\sum }}e_{jy}*\omega_{jy}=-\overset{n}{\underset{i=1}{\sum }}\frac{v_{jy}}{\sum_{1}^{n}v_{jy}}*\left(1-\overset{n}{\underset{i=1}{\prod }}\left(0.8+\frac{y_{ji}-y_{jmid}}{y_{jhigh}-y_{jlow}}\right)\right)\ln\overset{n}{\underset{i=1}{\prod }}\left(0.8+\frac{y_{ji}-y_{jmid}}{y_{jhigh}-y_{jlow}}\right)$$

Considering the weight ratio among subsystems, the indicator weight is adjusted according to the evaluation value of ship informatization degree and route safety value [42]. The evaluation value of informatization degree is used for the distribution of system strength, and the route safety value is adjusted as a whole. Thus, the total effectiveness can be obtained as follows:(19)$$E_{total}=\theta \overset{m}{\underset{j=1}{\sum }}E_{j}*f_{j}\left(t\right),\;\theta =\frac{r}{L}$$
where $f_{j}\left(t\right)$ represents the function of the degree of ship informatization with the age of the ship, θ represents the safety value of the route, r represents the number of annual accidents, and r represents the total annual route flow.

Combined with Equation (21), the total effectiveness value based on the performance characteristic quantity can be obtained as follows:(20)$$E_{total}=-\frac{r}{L}\overset{m}{\underset{j=1}{\sum }}\overset{n}{\underset{i=1}{\sum }}\frac{f_{j}\left(t\right)*v_{jy}}{\sum_{1}^{n}v_{jy}}\left(1-\overset{n}{\underset{i=a}{\prod }}\left(0.8+\frac{y_{ji}-y_{jmid}}{y_{jhigh}-y_{jlow}}\right)\right)*\ln\overset{n}{\underset{i=1}{\prod }}\left(0.8+\frac{y_{ji}-y_{jmid}}{y_{jhigh}-y_{jlow}}\right)$$

## 4. Case Study

### 4.1. Case Selection and Results Output

Accurate scenario definition is the basic guarantee for the evaluation preparation of system effectiveness. Considering that the research object is the sailing state of the ship, the verification scenarios need to be restricted before case selection and indicator data collection.

Generally, a ship’s transportation cycle is the process from the previous port to the next port, including loading, unberthing, leaving port, sailing, entering port, berthing, and unloading [43]. Since this study mainly focuses on the effectiveness evaluation of the risk mitigation system for navigating ships, the scenario is limited to the processes of unberthing, leaving port, sailing, entering port, and berthing, as shown in Figure 11.

In Figure 11, it can be seen that ship navigational tasks can be reorganized into three main stages:Ship departure. In the departure stage, ships go through the process of unberthing and leaving port. Ship risk mitigation mainly involves unberthing safety, sailing in narrow waters, and route planning.Ship sailing. In the sailing stage, ships go through the process of multiple navigation areas and intersection navigation. Ship risk mitigation mainly involves the unberthing safety of navigation monitoring, ship–shore communications, and collision avoidance decisions.Ship arrival. In the arrival stage, ships go through the process of entering ports and berthing. Ship risk mitigation mainly involves the unberthing safety of sailing in narrow water navigation monitoring, ship–shore collaboration, and berthing safety.

Based on the ship navigational information platform in a test ship named Yukun, we carried out an exploratory test in an SRMS to verify whether the methodology could accurately evaluate the effectiveness of each subsystem proposed in Section 3.1. The specific parameters of test ship named Yukun is shown in Table 7 below. This test was based on the information collected from an actual ship. The main data types we collected were based on navigation records to complete the risk mitigation tasks, and then we tested and verified them through changes in interaction frequency, volume, and complexity. We took the data as the input and the indicator results under the test as the output. Then, we monitored the effectiveness change of the system and the performance difference under different indicator data. To improve the data consistency, the same set of data measurement standards was used to collect data from all tests.

Considering that the research involved 3 subsystems including 36 capabilities and 82 groups of influencing elements, we selected one indicator of the 82 elements to analyse the model results for the verification of degradation methodology and 36 capabilities to analyse the entropy of capability calculation process for the verification of effectiveness evaluation methodology. The test data and corresponding effectiveness evaluation values are shown in Appendix A. Based on the inspection practice for the risk mitigation system on board of Yukun, five points are selected chronologically to measure the degradation of the risk mitigation system. The measured values for the indicator of “basic personnel support capacity” in human defence subsystem at these five points are exampled in Appendix A by multiple cycle test [44], and the measured results are shown in Figure 12, based on which the statistical analysis data are summarized in Table 8. 

In addition, based on the effectiveness measurement method, we calculated the statistics of relevant indicators under the three subsystems of human defence, physical defence, and technical defence, as shown in Table 9.

### 4.2. Degradation Regulation Analysis

#### 4.2.1. Volatility Analysis

Anderson darling test is a non-parametric test method to test whether the collected data obey a certain distribution (such as normal distribution, exponential distribution, Weber distribution, etc.) or not. For a given data and distribution, the better you sum up, the smaller the value will be. A certain given value can be used to test whether the data comes from the given distribution or not. For instance, if the given value is less than a certain threshold value (e.g., 0.05), then the original hypothesis would be rejected, and the data does not obey the distribution. For the data at points A–E, the Anderson–Darling test was used to detect the distribution form, and the test results are shown in Table 10 [45].

In Table 10, based on the comparison between statistical values and critical values, it is found that the data obeyed the normal distribution with a confidence of 95% but did not obey the exponential distribution.

In Table 8, the test results at five points were $E_{j}\left(A\right)=0.863755$, $E_{j}\left(B\right)=0.788881$, $E_{j}\left(C\right)=0.653696$, $E_{j}\left(D\right)=0.513926$, and $E_{j}\left(E\right)=0.431334$. As shown in Figure 13, the results are in accordance with the regulation of a gradual decline.

In Figure 13, it can be seen that the measured value of each point has a large fluctuation, so it is verified that the expected average value can effectively improve the accuracy of measurement compared with the single measurement.

Additionally, Figure 13 shows that the value of the effectiveness characteristic parameter gradually degenerates as the system life cycle develops. The results are consistent with the actual capability degradation trend and verify the effectiveness and applicability of the methodology.

Based on the standard deviation (std) of the expected values in Table 8 and the regional dispersion degree in Figure 14, it can be determined that with the capability degradation of the element, the standard deviation presents a gradual upwards trend, and the upwards trend becomes faster in the later stage. The results show that the system performance fluctuates greatly after degradation. This is also consistent with the increasing uncertainty of system capability over time.

#### 4.2.2. Degradation Trend Analysis

To master the effectiveness degradation regulation of SRMS in the whole life cycle, we divided 250 measurement points according to the time series and expressed them in the form of a scatter diagram. The data were fitted by exponential, logarithmic, and Weibull curves, as shown in Figure 14.

The fitting forms of three common curves are tested on the data, and the results are shown in Table 11. In Table 11, it is found that compared with exponential curve and logarithmic curve fitting, the data were subjected to the Weibull curve.

Through calculation by the Python flatform, the fitting functions of the curves were as follows:(21)$$\mathrm{Weibull}:\;E\left(t\right)=\frac{\kappa }{\lambda }{\left(\frac{t}{\lambda }\right)}^{\kappa -1}\times e^{-{\left(\frac{t}{\lambda }\right)}^{\kappa }},\;\kappa =99.49,\;\lambda =1.7$$
(22)$$\mathrm{Expponential}:\;E\left(t\right)=ae^{\left(bx\right)}+c,\;a=0.91,\;b=-0.004,\;c=0.1$$
(23)$$\mathrm{Logarithmic}:\;E\left(t\right)=alog(x-b)+c,a=-0.2,b=50,c=1.48$$

In Figure 14, it can be determined that the fitting curve shows a downwards trend with time, which also corresponds to the above five groups of measured points, indicating that the curve conforms to the corresponding degradation regulation. Additionally, it is found that the degradation curve in the first 25% of the life cycle belongs to a relatively stable stage, the degradation rate gradually increases at 25~50% of the life cycle, decreases at 50–75% of the life cycle, and the system performance reaches a relatively low stable state at 75–85% of the life cycle. These results are consistent with the phenomenon that the SRMS can operate but is not efficient in reality. A value at 75–85% of the life cycle ($E_{j}\left(250\right)\approx 0.4$) that is not 0 also shows that there is a large error in the traditional use of a detection value of 0 to represent the failure of the risk mitigation system [46]. By comparing the fitting results of the three common curves in Figure 14, it can be seen that the curve fitting can better express the effectiveness degradation regulation of the SRMS.

### 4.3. Evaluation Accuracy Analysis

Based on the data collection and sorting of the risk mitigation indicator, the effectiveness measurement algorithm based on the entropy of capability was compared with the traditional accident probability algorithm to verify the accuracy of the effectiveness evaluation model. Considering the uncertainty of using the subjective scoring method in many traditional accident probability measurement algorithms, the prior experience of historical data is selected as the basis of the weight ratio setting to reduce the impact of subjective evaluation results on the analysis results.

#### 4.3.1. Traditional Effectiveness Evaluation Method Based on the Accident Probability Algorithm

(1)Priori data generation

Based on the common framework of the ship risk mitigation system, we collected accident statistics from the HIS sea–web database in the past 10 years, and we conducted statistical analysis on the main types of accidents and the causes of accidents, as shown in Figure 15.

Combining specific accident cause classification and statistical reports and following the common five types of ship risk mitigation, the cause statistical value table is constructed as in Table 12.

(2)Effectiveness calculation

Since the traditional accident probability algorithm is based on actual ship accidents, we compiled the maintenance records of the tested ship in the past 1 year and sorted out the relevant data, as shown in Table 13.

Based on the failure probability of each element, we use the weight ratio of a priori experience to calculate the effectiveness value as follows:(24)$$E=\overset{m}{\underset{j=1}{\sum }}\left(\overset{n}{\underset{i=1}{\prod }}\left(\frac{1-\frac{x_{ij}}{x_{ej}*m}\left(1-\delta \right)}{1-p_{ij}}\right)*\partial_{j}\right)$$
where $x_{ij}$ represents the ith causal indicator value of the jth risk mitigation subject, $x_{ej}$ represents the reference value of the jth risk mitigation subject, δ represents the sailing time correction factor, and $p_{ij}$ represents the a priori failure probability value of the ith cause indicator of the jth risk mitigation subject. $\partial_{j}$ represents the weight factor of the jth risk mitigation subject.

Based on the failure probability of each element, we calculated the effectiveness of the SRMS, as shown in Table 14.

#### 4.3.2. Effectiveness Evaluation Based on Entropy of Capability

Based on the data collection in the technical defence, human defence, and physical defence subsystems, using the effectiveness measurement methodology in Section 3.4, the corresponding subsystem and system effectiveness values can be obtained as shown in Table 15.

Then, based on the calculation standard of casualties and property loss of the international labour organization, the real navigational efficiency value of the ship is calculated. Combined with the actual sailing accident record data of the test ship in the past year, we conduct a comparative analysis of the two methods, as shown in Figure 16. 

Figure 16 shows that the deviation between the effectiveness value based on the entropy of capability and the actual value is approximately 4.93%. The deviation of the traditional effectiveness value based on accident probability from the actual value can reach 13.58%, which is much larger than the effectiveness based on the entropy of capability. Therefore, the effectiveness model based on the entropy of capability can more accurately represent the real state of ship risk mitigation, with an accuracy improvement of 9%.

Based on the above definition of performance in Section 2.1, The effectiveness defined in this research refers to the degree to which the vessel risk mitigation system has completed risk control, not to the operation efficiency of the system itself. From the statistical historical data, the traditional effectiveness calculation methodology is mainly based on accident statistics and can only be calculated after the accident occurred, ignoring the degradation process of the capability to cause the accident from quantitative change to qualitative change. Therefore, there is a certain lag in the effectiveness evaluation of an SRMS. In addition, the problem of non-accidental dangerous phenomena may not be collected. These reasons combined lead to the phenomenon of a large distortion of evaluation value. This method may lead to a certain risk of error in the understanding of risk mitigation status, which will seriously affect the safety of ship navigation. In contrast, the effectiveness model based on the entropy of capability proposed in this study is based on the measurement of the uncertainty of capability, and at the same time, the capability change caused by capability degradation over time is considered. Therefore, the effectiveness evaluation based on the entropy of capability is more accurate and applicable.

Figure 16 shows that the calculated effectiveness value based on the entropy of capability is also slightly larger than the true value. Through the decomposition of the model, it is found that since the complexity and uncertainty of the internal relationship in the SRMS may bring certain systematic failure, the real effectiveness will be slightly lower than the shown status. Therefore, this further confirms that the calculation method of the SRMS using entropy of capability is closer to the real state.

### 4.4. Model Comprehensive Analysis

From the comparative analysis in Section 4.2 and Section 4.3, the effectiveness evaluation model based on the entropy of capability has better advantages in coverage, timeliness, accuracy and operability. The evaluation framework of the SRMS based on the theoretical perspective of the comprehensive defence system is more scientific and can provide a better network architecture for accurately measuring the effectiveness of the SRMS. In addition, compared with the traditional ergodic indicator design, the architecture from the capability perspective is more systematic and coverage. From the volatility analysis of system capability degradation, the model considers and analyses the dynamic variability of system capability and improves the timeliness characteristics of the effectiveness value. Additionally, the construction of the entropy of capability in this study solves the problem that capability is difficult to measure directly. In addition, it constructs the corresponding effectiveness measurement methodology based on the risk mitigation tasks. Therefore, it improves the accuracy of the evaluation of the actual risk mitigation state and defines the operability of the improvement of risk mitigation capability.

However, since the model has less sample verification, its versatility needs to be further determined. In addition, the suddenness and complexity of ship safety risks are relatively serious, and the corresponding causal relationships need to be further studied.

## 5. Conclusions

Sustaining ship safety through navigational stability is highly reliant on the effectiveness of ship risk mitigation systems. To comprehensively consider the degradation characteristics of risk mitigation capability and the scenario attributes of the risk mitigation task, this study proposed an effectiveness evaluation model based on the entropy of capability construction and capability degradation analysis. Beginning with the transformation of the risk mitigation concept in the SRMS and referring to the three-dimensional integrated risk mitigation framework of comprehensive defence theory, human defence, physical defence and technical defence, the SRMS indicator system is established based on capability building, which breaks through the traditional evaluation method based on accident probability analysis. On this basis, a capability degradation analysis methodology was constructed to solve the problem of capability change over time. It realized the possibility of dynamic capability analysis of an SRMS and improved the accuracy of effectiveness evaluation of an SRMS. In addition, because the risk mitigation capability cannot be directly measured, the entropy of capability is constructed based on entropy theory. On this basis, an effectiveness evaluation model based on the entropy of capability was proposed. From the perspective of capability uncertainty measurement, the effectiveness evaluation of risk mitigation status in the SRMS was realized by the model proposed in the study.

The case study results show that using the entropy of capability can realize the accurate effectiveness evaluation of a risk mitigation system for navigating ships with an accuracy improvement of 9%, and the Weibull curve fitting is more consistent with the capability degradation regulation with a significance level of 2.5%. Our research provides a more accurate and comprehensive dynamic analysis method for the effectiveness evaluation of SRMSs. Instead of simple statistics of accident probability and substituting the measurement of capability indicators, it is a comprehensive analysis of the measurement of capability uncertainty and timing degradation. This framework, design based on capability building, ensures good applicability for evaluating the effectiveness of SRMSs on multilevel ships. The present study verified the proposed model for one case. For the application in the practical operation, the application procedure needs to be further standardized to facilitate the engineers’ utilization, and the test from the third party may be also required. Further research work for the present study would be focused on the practical application of the proposed model for different ship fleets, based on which, the standardized application procedure would be developed.

## Figures and Tables

**Figure 1 ijerph-19-09338-f001:**
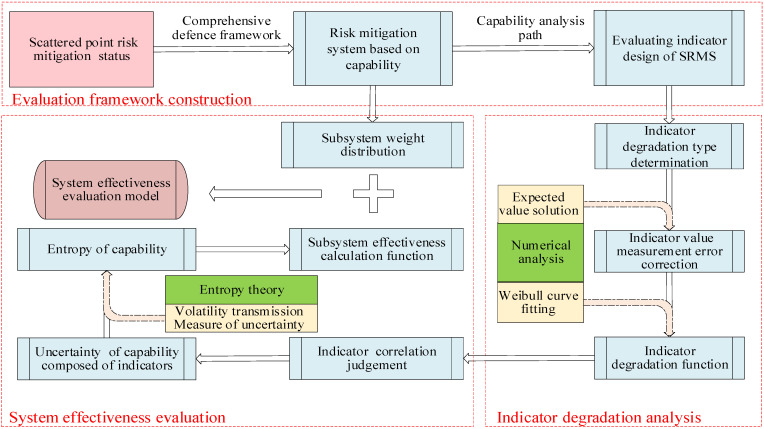
Schematic diagram of the effectiveness evaluation model based on entropy theory.

**Figure 2 ijerph-19-09338-f002:**
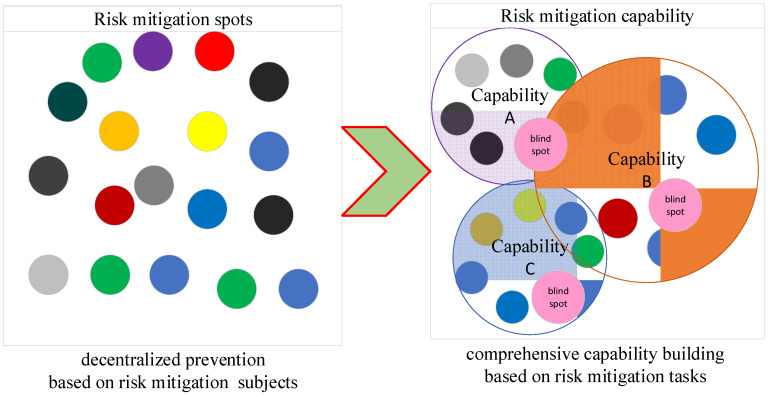
The comprehensive security system theory core principle.

**Figure 3 ijerph-19-09338-f003:**
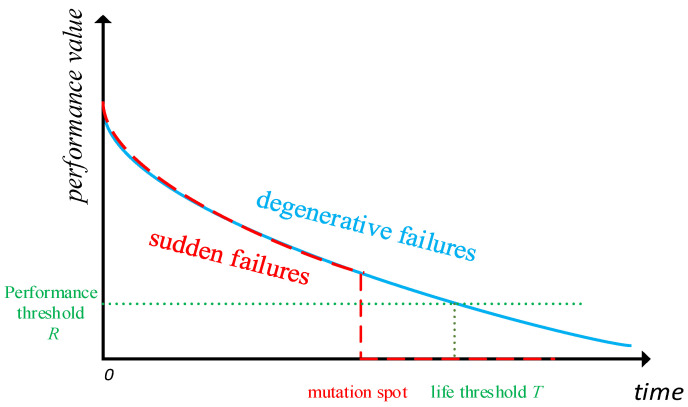
Performance change trends of sudden failures and degraded failures.

**Figure 4 ijerph-19-09338-f004:**
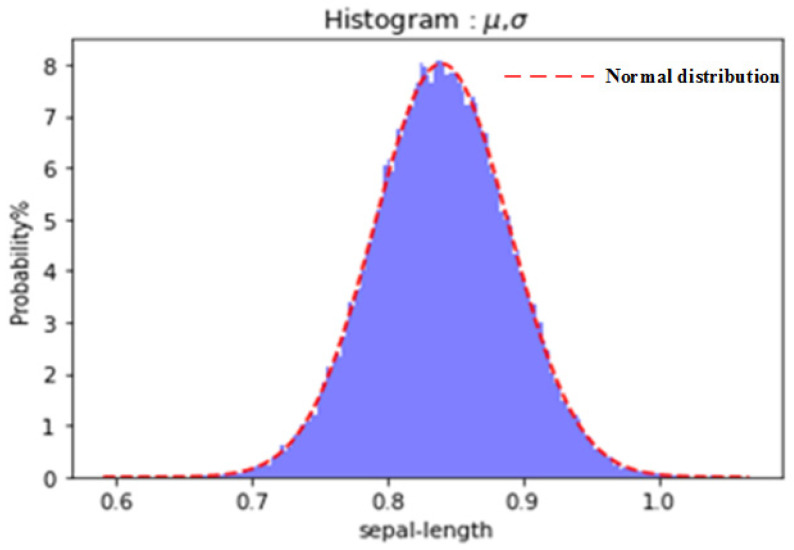
The distribution form selected for extracting the true value.

**Figure 5 ijerph-19-09338-f005:**
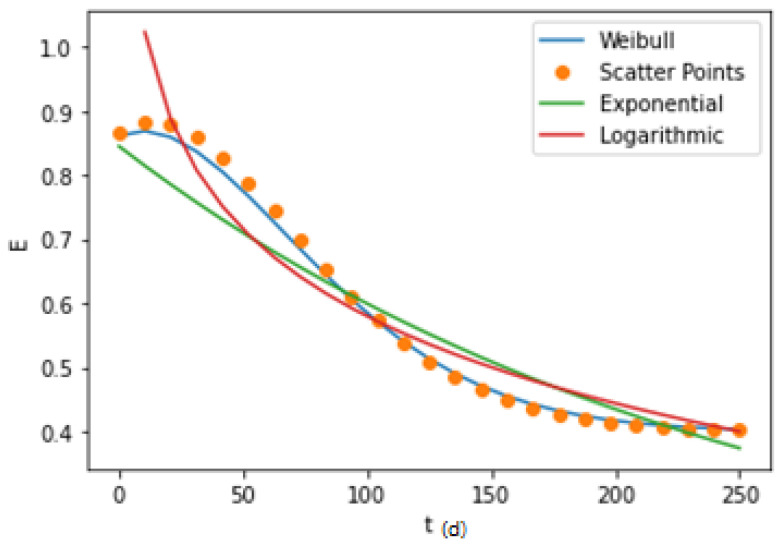
Comparison of various curve fitting forms of characteristic quantities.

**Figure 6 ijerph-19-09338-f006:**
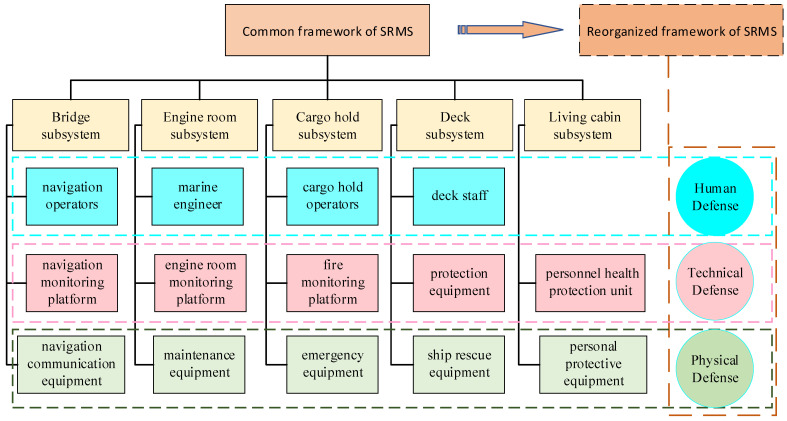
The common and reorganized framework of an SRMS.

**Figure 7 ijerph-19-09338-f007:**
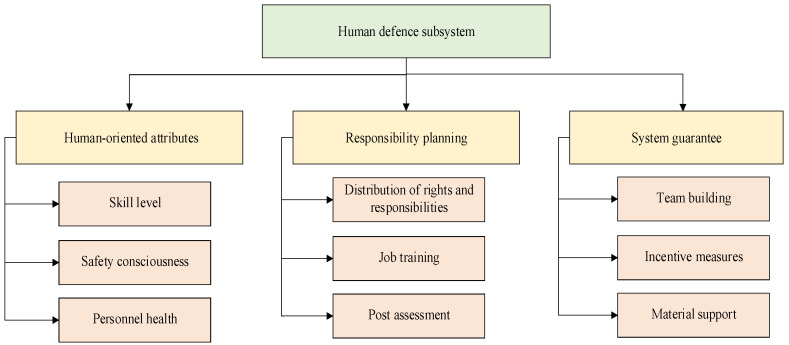
Measurement framework of the human defence system capability indicator.

**Figure 8 ijerph-19-09338-f008:**
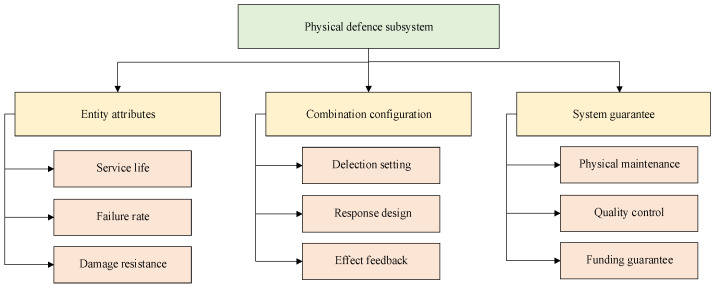
Measurement framework of the physical defence system capability indicator.

**Figure 9 ijerph-19-09338-f009:**
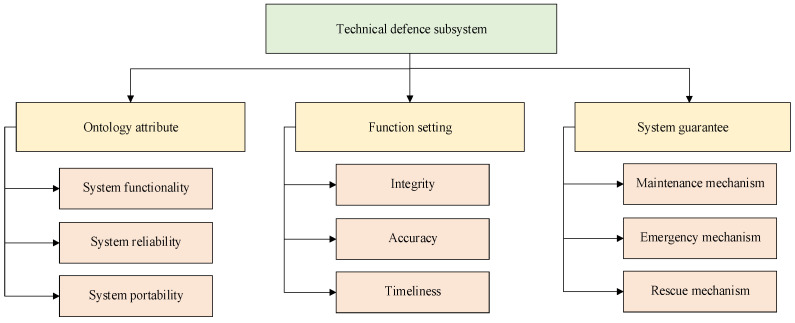
Measurement framework of the technical defence system capability indicator.

**Figure 10 ijerph-19-09338-f010:**
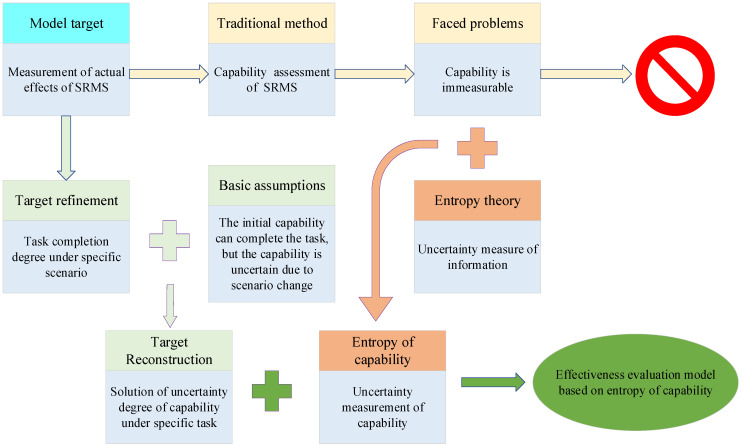
Derivation logic of effectiveness measurement based on entropy of capability.

**Figure 11 ijerph-19-09338-f011:**
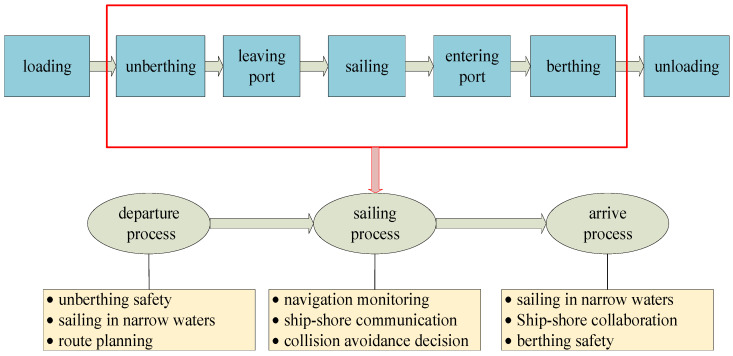
Scenario settings for the effectiveness evaluation of SRMS.

**Figure 12 ijerph-19-09338-f012:**
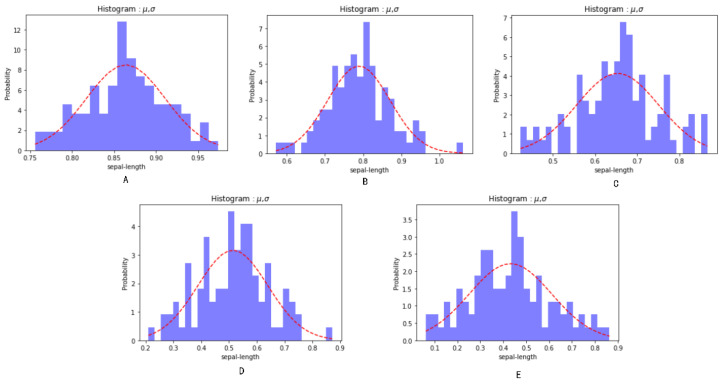
Measured results: distribution at points (**A**–**E**).

**Figure 13 ijerph-19-09338-f013:**
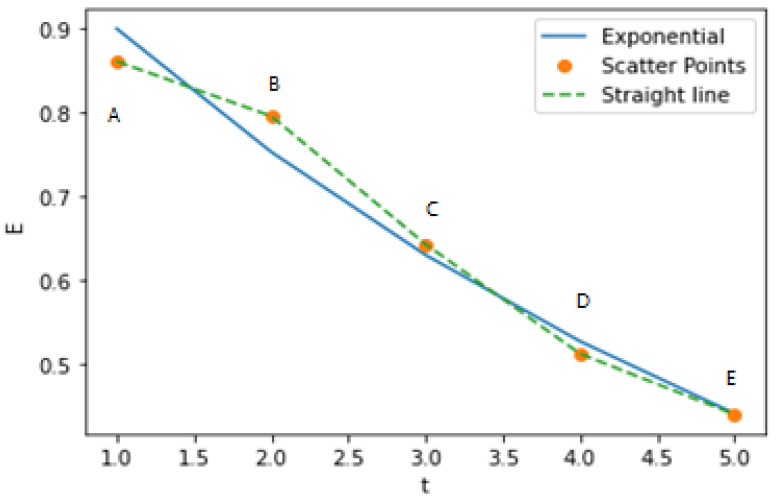
Degradation trend of the data at points A–E.

**Figure 14 ijerph-19-09338-f014:**
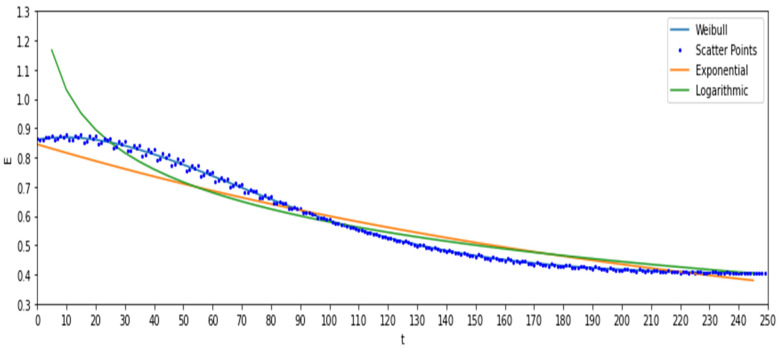
Fitting curves of the statistical analysis data at 250 measurement points.

**Figure 15 ijerph-19-09338-f015:**
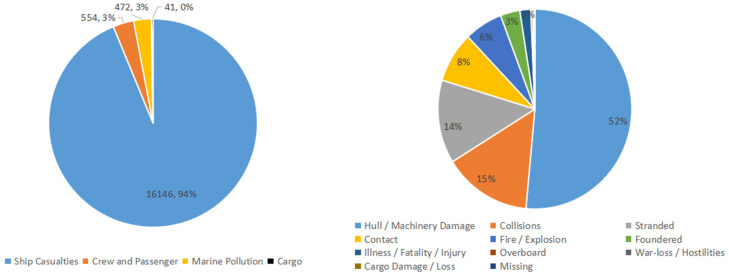
The main accident type and cause from the IHS database in 2011–2021.

**Figure 16 ijerph-19-09338-f016:**
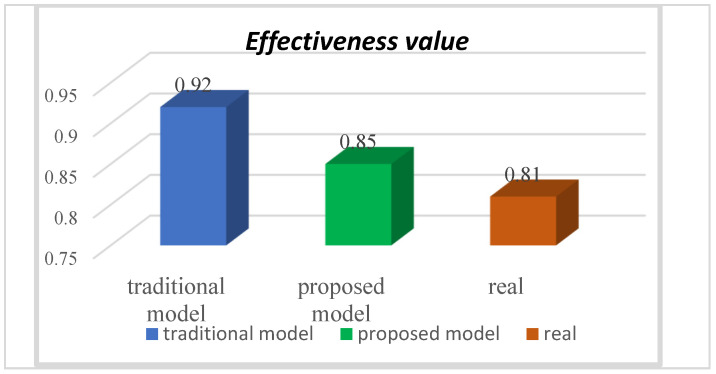
Comparison of the results from the two models.

**Table 1 ijerph-19-09338-t001:** Details of the human defence subsystem of the SRMS.

Composition	A collection of human-centred risk mitigation standards, measures and behaviours.
objective	From the perspective of human risk mitigation, fully mobilize people’s subjective initiative, timely identify, accurately judge and efficiently deal with potential safety risks to ensure the normal operation of risk mitigation subjects.
function	Based on people’s subjective initiative, give play to the role of independent decision-making, flexible interaction, and experience dependence of human defence.
characteristic	Advantage: strong individual dependence (experience, capability), strong autonomy (self-judgement), strong interaction (flexible), strong adaptability (changing with the environment).
	Disadvantage: poor uniformity (judgement criteria, operation mode), poor driver (unable to standardize the program), poor persistence (human characteristics).

**Table 2 ijerph-19-09338-t002:** Details of the physical defence subsystem of the SRMS.

Composition	A collection of hardware-based risk mitigation instruments, signs and facilities.
objective	From the perspective of physical risk mitigation, giving full play to the passive resistance of objects, reducing, delaying and avoiding possible risks, and then improving the effectiveness of water traffic risk mitigation.
function	Based on the characteristics of a physical object, playing the role of barrier, identification, and damage resistance of physical prevention.
characteristic	Advantage: good damage resistance (compared to humans), high permanence (24 h on duty), excellent economy (compared with personal injury), strong replaceability (replaceable after damage), strong objectivity (fixed attributes).
	Disadvantage: poor autonomy (no self-awareness), poor flexibility (almost no interaction), poor lifting performance (cured performance).

**Table 3 ijerph-19-09338-t003:** Details of the technical defence subsystem of the SRMS.

Composition	A collection of technology-based risk mitigation methods, application software, and integrated systems.
objective	From the perspective of technical risk mitigation, giving full play to the effectiveness of technical activities, using information technology to discover, analyse, and deal with potential risks, then improving the effectiveness of water traffic risk mitigation.
function	Based on the inherent characteristics of technology, playing the role of identification, early warning, analysis, and monitoring in technical prevention.
characteristic	Advantage: strong methodology (with technical support), strong adaptability (fast perception speed), strong interactivity (link people and things), high integration (diversified constituent elements), good relevance and responsiveness (completely from actual demand).
	Disadvantage: high external dependence (cannot run alone), high technical threshold (not easy to achieve), faster stacking speed (as demand changes).

**Table 4 ijerph-19-09338-t004:** Capability indicators of human defence.

Basic Personnel Support Capability	The Certificate Status
	Ship Manning Situation
Stability of personnel on board	average time on board
	proportion of leader layer in half a year
	proportion of operation layer in half a year
	proportion of support layer in half a year
Familiarity of decision-makers with ships	the ship’s captain continuous on-board time
	the ship’s chief mate continuous on-board time
	average cumulative time of officers on similar ships
Risk mitigation exercise situation	completeness of types of ship exercises
	frequency of key project exercises
	record times of basic exercise training
Working language on board	proportion of native language crew
	proportion of crew nationality differentiation
Crew safety training	average annual class hours of organizations
	average annual training hours for crew enterprises
Crew handover situation	proportion of crew handover records
Implementation of pre shift meeting system	pre shift meeting record
	statistics of on-board operation accidents
Discussion on on-board safety risk events	number of participants
	frequency of discussion
Health status of on-board personnel	frequency of crew medical examination
	proportion of chronic occupational diseases
	frequency of psychological relief
Operation conditions on board	continuous monitoring under closed operation
	frequency of on-board operation

**Table 5 ijerph-19-09338-t005:** Capability indicators of physical defence.

Anti-piracy capability	safe house
	frequency of safety house inspections
Mobile firefighting capability	quota quantity
	instrument pressure (bar)
	inspection cycle
Closed space gas detection capability	alarm concentration
	number of false-positives
Video collection capability	definition (image resolution)
	signal-to-noise ratio (dB)
Bilge emergency pump discharge capability	lift (m)
	flow (m^3^/h)
	cavitation indicator
Safety warning capability	vent prompt bar
	smoking warning signs
	warehouse warning signs
Safe operation support capability	tag and lock off
Fuel safety protection capability	inspection frequency of quick closing valve
Personal protective equipment configuration capability	protective rope
	protective cap
	protective clothing
	gas protection equipment
	protective earplugs
Fixed fire extinguishing capability	trigger response value (mg/l)
	number of false-positives (times/month)
	gas emission rate (l/min)
Ship self-rescue capability	number of lifeboats
	number of life rafts
	number of lifebuoys
	number of life jackets
Fire isolation capability	airtightness of fire door
	alarm device
Water inlet alarm capability	alarm value of water inlet (mm)
	number of false-positives

**Table 6 ijerph-19-09338-t006:** Capability indicators of technical defence.

Ship’s automatic navigation capability	electronic chart update (times/month)
	GPS accuracy
Bridge information monitoring capability	coverage of monitoring indicators
	effective information fusion rate
Radar monitoring capability	determination accuracy
	anti-interference rate
Emergency communication capability	information fidelity
	channel capability (kb)
	communication delay (ms)
Route abnormal alarm capability	alarm value (deviation degree)
	number of alarms (times/day)
Automatic collision avoidance capability	accuracy of generation
	probability of scheme adoption
Meteorological monitoring capability	accuracy within 1 year
Fault tolerance of risk alarm system	fault tolerance degree
Network protection capability	protection software and hardware
	network paralysis response plan
Video surveillance coverage capability	coverage
Unsafe behaviour recognition capability	intelligent recognition rate
Ship–shore cooperative monitoring capability	VTS visibility in the jurisdiction
	visualization degree of remote sensing information
	sum of GNSS delay error and inherent error (ms)
	LRIT information protection mechanism
	GMDSS false alarm rate

**Table 7 ijerph-19-09338-t007:** The specific parameters of test ship named Yukun.

Total Length	116 m	Gross Tonnage	6000 t
width	18 m	speed	18 nm/h
depth	8.35 m	voyage	10,000 nm
design draft	5.4 m	construction date	2008

**Table 8 ijerph-19-09338-t008:** Statistical analysis data at five points of the proposed life cycle.

	A	B	C	D	E
Count	100	100	100	100	100
Mean	0.863755	0.788881	0.653696	0.513926	0.431334
Std	0.054391	0.081878	0.097234	0.126745	0.180557
Min	0.75591	0.572405	0.424012	0.210193	0.061739
25%	0.81945	0.737842	0.59573	0.421013	0.321971
50%	0.862128	0.78775	0.660608	0.518999	0.434303
75%	0.899203	0.832973	0.707531	0.588304	0.539868
Max	0.974195	1.060506	0.86603	0.87106	0.863276

**Table 9 ijerph-19-09338-t009:** Statistical analysis data for the 36 capability indicators under the three subsystems.

	Capability Indicators	Characteristic Value $q_{y}$	Improved Characteristic Value $e_{y}$	Loss Factor $\omega_{y}$	Entropy of Capability $I_{y}$	Subsystem Effectiveness $E_{j}$	Informatization Degree $f_{j}\left(t\right)$	Route Safety Factor θ	System Effectiveness $E_{total}$
physical defence	Anti-piracy capability	0.8000	1.288	1.92%	0.0248	0.9421	39%	0.96	0.8485
	Mobile firefighting capability	0.4705	0.299	3.85%	0.0115				
	Closed space gas detection capability	0.8333	1.493	5.77%	0.0862				
	Video collection capability	0.7680	1.122	7.70%	0.0864				
	Bilge emergency pump discharge capability	0.7147	0.896	9.62%	0.0862				
	Safety warning capability	1.0000	2.000	11.54%	0.2309				
	Safe operation support capability	1.0000	2.000	13.47%	0.2694				
	Fuel safety protection capability	0.9333	2.528	15.39%	0.3891				
	Personal protective equipment configuration capability	1.0000	2.000	17.32%	0.3463				
	Fixed fire extinguishing capability	0.5457	0.431	1.92%	0.0083				
	Ship self-rescue capability	0.7680	1.122	1.92%	0.0216				
	Fire isolation capability	0.8000	1.288	3.85%	0.0495				
	Water inlet alarm capability	0.5514	0.442	5.77%	0.0255				
technical defence	Ship’s automatic navigation capability	0.6267	0.617	6.25%	0.0386	0.8294	27%		
	Bridge information monitoring capability	0.6618	0.717	7.81%	0.0560				
	Radar monitoring capability	0.6333	0.635	9.38%	0.0596				
	Emergency communication capability	0.7467	1.025	10.94%	0.1121				
	Automatic collision avoidance capability	0.4267	0.237	12.50%	0.0297				
	Meteorological monitoring capability	0.5973	0.5434	14.06%	0.0764				
	Fault tolerance of risk alarm system	0.8360	1.5111	15.63%	0.2361				
	Network protection capability	0.8000	1.2876	1.56%	0.0201				
	Video surveillance coverage capability	1.0000	2.0000	3.13%	0.0625				
	Unsafe behaviour recognition capability	0.8360	1.5111	4.69%	0.0708				
	Ship–shore cooperative monitoring capability	0.8100	1.3452	6.25%	0.0841				
	Ship’s automatic navigation capability	0.1960	0.0428	7.81%	0.0033				
human defence	Basic personnel support capability	1.0000	2.0000	8.86%	0.1772	0.8604	34%		
	Familiarity of decision-makers with ships	0.1851	0.0379	10.34%	0.0039				
	Risk mitigation Exercise situation	0.4642	0.2897	11.82%	0.0342				
	Working language on board	0.3698	0.1707	13.29%	0.0227				
	Crew safety training	0.4949	0.3381	1.48%	0.0050				
	Crew handover situation	0.7200	0.9165	2.95%	0.0271				
	Implementation of pre-shift meeting system	0.8000	1.2876	4.43%	0.0570				
	Discussion on on-board safety risk events	0.8000	1.2876	5.91%	0.0761				
	Health status of on-board personnel	0.8000	1.2876	7.38%	0.0951				
	Operation conditions on board	0.2880	0.0978	8.86%	0.0087				
	Stability of personnel on board	0.5333	0.4065	23.63%	0.0961				

**Table 10 ijerph-19-09338-t010:** Distribution pattern test results based on Anderson–Darling test.

Distribution Form	Feature	Flag (0.05)	Statistic	Critical Values	Signification Level
Normal distribution	A	+	0.262144011	[0.555 0.632 0.759 0.885 1.053]	[15. 10. 5. 2.5 1.]
	B	+	0.240336575	[0.555 0.632 0.759 0.885 1.053]	[15. 10. 5. 2.5 1.]
	C	+	0.447665853	[0.555 0.632 0.759 0.885 1.053]	[15. 10. 5. 2.5 1.]
	D	+	0.341666108	[0.555 0.632 0.759 0.885 1.053]	[15. 10. 5. 2.5 1.]
	E	+	0.30482838	[0.555 0.632 0.759 0.885 1.053]	[15. 10. 5. 2.5 1.]
Exponential Distribution	A	-	41.12834048	[0.917 1.072 1.333 1.596 1.945]	[15. 10. 5. 2.5 1.]
	B	-	37.26900374	[0.917 1.072 1.333 1.596 1.945]	[15. 10. 5. 2.5 1.]
	C	-	33.61289887	[0.917 1.072 1.333 1.596 1.945]	[15. 10. 5. 2.5 1.]
	D	-	26.24084953	[0.917 1.072 1.333 1.596 1.945]	[15. 10. 5. 2.5 1.]
	E	-	14.97244258	[0.917 1.072 1.333 1.596 1.945]	[15. 10. 5. 2.5 1.]

**Table 11 ijerph-19-09338-t011:** Test results of three curve shape fitting.

Feature	Flag	Statistic	Critical Values	Signification Level
Exponential	-	61.50365	[0.921 1.075 1.338 1.602 1.952]	[15. 10. 5. 2.5 1.]
Logarithmic	-	12.30670	[0.426 0.562 0.659 0.768 0.905 1.009]	[25. 10. 5. 2.5 1. 0.5]
Weibull	+	0.447665	[0.362 0.395 0.427 0.462 0.506]	[15. 10. 5. 2.5 1.]

**Table 12 ijerph-19-09338-t012:** Traditional accident statistical data of the IHS database in 2011–2021.

Subject	Distribution Type	Number	Total	All	Weights $p_{ij}$
bridge subsystem	navigation operators	2694	5786	16,146	16.69%
	navigation monitoring and warning	2076			12.86%
	navigation communication equipment	1016			6.29%
engine room subsystem	marine engineer	1519	3461		9.41%
	engine room monitoring platform	777			4.81%
	maintenance equipment	1165			7.22%
cargo hold subsystem	operators	1032	2643		6.39%
	monitoring system	1032			6.39%
	emergency equipment	579			3.59%
deck subsystem	staff	2321	3256		14.37%
	protection equipment	381			2.36%
	ship rescue equipment	554			3.43%
living cabin subsystem	personal protective equipment	536	1000		3.32%
	personnel health protection unit	464			2.88%

**Table 13 ijerph-19-09338-t013:** Statistical data of maintenance records of the tested ship in the past 1 year.

Subject	Distribution Type	Failure $x_{ij}$	Criterion $x_{ej}$	Sailing Times m	Sailing Time Correction Factor δ	Probability
bridge subsystem	navigation operators	16	6	55	100/270	0.9695
	navigation monitoring and warning	7	4			0.9800
	navigation communication equipment	4	2			0.9771
engine room subsystem	marine engineer	12	4			0.9657
	engine room monitoring platform	9	2			0.9485
	maintenance equipment	5	3			0.9809
cargo hold subsystem	operators	6	4			0.9828
	monitoring system	13	4			0.9628
	emergency equipment	6	2			0.9657
deck subsystem	staff	12	6			0.9771
	protection equipment	1	1			0.9828
	ship rescue equipment	3	1			0.9657
living cabin subsystem	personal protective equipment	3	1			0.9657
	personnel health protection unit	2	1			0.9771

**Table 14 ijerph-19-09338-t014:** Effectiveness value based on the traditional accident probability algorithm.

	Subject	Effectiveness	Weights $\partial_{j}$	Total E
common framework of SRMS based on spatial distribution	bridge subsystem	0.9283	35.84%	0.9202
	engine room subsystem	0.8984	21.44%	
	cargo hold subsystem	0.9138	16.37%	
	deck subsystem	0.9273	20.17%	
	living cabin subsystem	0.9435	6.19%	

**Table 15 ijerph-19-09338-t015:** Effectiveness value based on the entropy of capability construction.

	Subject	Effectiveness	Total
SRMS based on capability construction	human defence subsystem	0.8604	0.8485
	physical defence subsystem	0.9421	
	technical defence subsystem	0.8294	

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
