# Peer review of "A Model to Evaluate the Effectiveness of the Maritime Shipping Risk Mitigation System by Entropy-Based Capability Degradation Analysis"

_ijerph, 2022, doi:10.3390/ijerph19159338_

Round 1
Reviewer 1 Report
The paper investigates the evaluation of the effectiveness of maritime risk mitigation system under high uncertainties. The idea of Entropy is introduced, which can enhance the accuracy of the final results. The idea is new and attractive. The paper is well written.
One general impression on the paper is that the model is very large and contains rather many factors that are related with risk mitigation measures. It may need much more work to do if been applied to other relevant shipping safety systems.
Section 2.2 contains a lot of basic methodologies that readers may be familiar with. It is recommended to make them more conscise.
The data in appendix A is not easy to understand. It only contains pure data without explaination. It should be explained in more detail.
Author Response
Dear reviewer,
Thank you for your comments concerning our manuscript entitled “Barriers involved in the safety management systems: A systematic review of literature”. The authors are very appreciative of the valuable comments and recommendations provided for the improvement of the above-referenced manuscript. We have studied comments carefully and have made correction which we hope meet with approval. Revised portion are marked in red in the manuscript. And the responses to your comments can be found in the attached file.
Once again, thank you very much for your comments and suggestions.

Reviewer 2 Report
The risk mitigation system (RMS) has been investigated in this research paper. The effectiveness evaluation method has been proposed for the risk mitigation system based on the Entropy-based Capability Degradation Analysis through the physical defense, the technical defense, and the human defense. This proposed methodology is interesting in the field of maritime risk management as well as the safety science. However, there are some weakness points that the authors should address them to improve their research paper.
- The scientific writing in English should be improved in this research. For example:
Page 2, lines 63-66: Based on the above analysis, we believe that the current SRMS is a complex multidimensional mitigation system composed of human-ship interactions, ship-shore interactions, and ship-ship interactions, …
- Page 2, lines 71-75, “However, with the widespread application of advanced information technology in the shipping industry, the level of intelligence in risk mitigation measures is increasing, and the internal structures and interaction activities are becoming increasingly complex and uncertain.” This statement is confused. Kindly re-write this sentence clearly and explain it. The proof should be provided to make clear this statement.
- In the methodology part, the degradation analysis theory should be provided in this research.
- The contribution of this research as well as the novelty points must be provided both the safety science and the risk management in maritime transportation.
- In figure 4, the explanation of curves should be indicated in this figure. Additionally, the values of probability are very low under 10% corresponding 0.1.
- In figure 5, the unit of time should be provided.
- In this research, the authors had used the database from the certain ship namely Yukun. The authors had not stated any the specific parameters of this vessel. The authors should provide clearly in this research paper.
- Page 22, line 775, there is one incorrect space. “Table 8 statistical analysis data at five points of the proposed life cycle.” It is not table 8.
- The table 8 is lacked in this research paper. Additionally, the different features of each point (A-E) in figure 12 should be indicated clearly.
- On the other hand, the authors should indicate clearly the scientific basis or which theory that that the authors had classified the database from the experimental vessel into five points.
- The Anderson-Darling testing method should be introduced briefly.
- In figure 13, the degradation trend of points A-E had been presented. However, the points A-E could not been provided in this figure.
- In figures 15 and table 12, the interval time of 10 years should be indicated clearly.
- In figure 16, the calculation of effectiveness values has been conducted. However, the collected results have been indicated that the effectiveness value of the proposed model is lower than the traditional model. The reliable degree of proposed method is inadequate in this research.
- The acknowledgement should be provided in this research.
- The reference sources must be cited sufficiently in this research paper.
Author Response

(The authors gave the same response as above.)

Round 2
Reviewer 2 Report
The revised paper is better now and it could be accepted to publish on the International Journal of Environmental Research and Public Health.